# Evaluation of Heat Treatment Parameters on Microstructure and Hardness Properties of High-Speed Selective Laser Melted Ti6Al4V

**Paul Lekoadi** [1,2,*]**, Monnamme Tlotleng** [1,3]**, Kofi Annan** [4]**, Nthabiseng Maledi** [2] **and Bathusile Masina** [1,*]

1   Council for Scientific Industrial Research (CSIR), Manufacturing, National Laser Centre,
    Laser Enabled Manufacturing Group, P.O. Box 395, Pretoria 0001, South Africa; MTlotleng@csir.co.za
2   School of Chemical and Metallurgical Engineering, University of Witwatersrand, Private Bag 3, Wits,
    Johannesburg 2050, South Africa; Nthabiseng.maledi@wits.ac.za
3   Department of Mechanical Engineering Science, University of Johannesburg, Auckland Park Campus,
    Johannesburg 2012, South Africa
4   Department of Material Science and Metallurgical Engineering, University of Pretoria,
    Mineral Science Building, Pretoria 0001, South Africa; Kofi.annan@up.ac.za
*   Correspondence: PLekoadi@csir.co.za (P.L.); BMasina@csir.co.za (B.M.);
    Tel.: +27-012-841-4619 (P.L.); +27-012-841-2121 (B.M.)

**Abstract:** This study presents the investigation on how heat treatment parameters, which are temperature, cooling method, and residence time, influence the microstructural and hardness properties of Ti6Al4V components produced on Ti6Al4V substrate using high speed selective laser melting technique. Heat treatment was performed on the produced samples before they were characterized for microstructure and hardness. The microstructure of the as-built sample contained large columnar β-grains that were filled with martensite α' phase and had a high hardness of 383 ± 13 HV. At 1000 °C and residence time of maximum 4 h, better heat treatment parameters were seen for the selective laser melting (SLM) produced Ti6Al4V sample since an improved lamellar α + β microstructure was obtained at this condition. This microstructure is known to have improved tensile properties.

**Keywords:** selective laser melting; heat treatment; temperature; cooling method; residence time; microstructure; hardness





## 1. Introduction

Selective laser melting (SLM) is a type of powder bed fusion (PBF) additive manufacturing (AM) technology that creates near net 3D shapes by using a laser as an energy source to melt a metallic powder layer-by-layer according to control aided design (CAD) models [1–3]. The fundamentals of SLM processing are discussed elsewhere [3]. SLM is regarded as a revolutionary manufacturing technique since it is precise when printing parts with complex geometries and high dimensional accuracy [4,5]. Furthermore, SLM is famously known to process various metallic powders and their alloys [6]. Titanium alloy, Ti6Al4V (also abbreviated as Ti64), has properties that are promising for high-end structures; hence, it is regard best for biomedical and aerospace engineering applications [7]. Ti6Al4V can be manufactured with ease into artifact using conventional processes [2,3]; however, in recent years, the developments of using SLM to produce Ti6Al4V components have been a subject of interest [7].

Ti6Al4V is composed of two primary stable phases commonly known as alpha (α) and beta (β). However, due to high heating and cooling rates that are associated with the SLM process, the microstructure of SLM produced Ti6Al4V parts is characterized by a single acicular sharp martensitic α' phase [8,9]. The martensite α phase is undesired for many applications, including aerospace, since it is associated with microstructural inhomogeneity, high hardness, low tensile elongation, and high residual stresses [10–12]. This implies that SLM produced Ti6Al4V components requires post-processing techniques, such as heat treatment, before they can be used.

Heat treatment is typically applied to SLM produced Ti6Al4V components since it can, in addition to relieving a print of the inherent internal thermal stresses, dissolve and transform the metastable $\alpha'$ phase into a desired microstructure with improved mechanical properties [13]. In order to achieve a fully transformed and superior two phase ($\alpha + \beta$) Ti6Al4V through post-heat treatment, it is critical to understand the role of heat treatment parameters, such as temperature, cooling method, and residence time, individually and in collective, on the improvement of microstructure and mechanical properties of Ti6Al4V.

Wu et al. [14] using laser power and scanning speed of 95 W and 900 mm/s and layer thickness of 25μm, were able to SLM 3D print Ti6Al4V cubes, which they used to for heat treatment studies. Specimens were heat-treated to different temperatures that varied from 300–1020 °C, followed with water quenching (WQ). The microstructure of the as-printed sample was characterized with a columnar prior $\beta$ grains that were filled with martensite $\alpha'$ phase. The columnar grains were associated with epitaxial growth of the $\beta$ phase while the martensite $\alpha'$ phase was associated with high cooling rates that are prevalent during SLM processing. A similar SLM microstructure has been reported by several other independent authors [3,12,15]. Post-heat treatment, at 600 °C, Wu et al. [14] did not observe any phase transformation or microstructural change. A change in microstructure was reported in the samples that water quenched after heat treatment in the temperature range between 750–950 °C.

In a different, but related, study, Chicos et al. [15] manufactured cylindrically shaped Ti6Al4V specimens from an SLM machine using laser power of 200 W, layer thickness of 25 μm, hatch spacing of 0.095 m, and scanning speed in the range 1200–1500 mm/s and performed heat treatment on the produced specimens. Samples were heat treated to 705 °C, 850 °C, 940 °C, 1015 °C, and 1050 °C for 2 h in a solar furnace at heating rate of 60 °C and gas flow rate of 200 L/min. Two methods of cooling (furnace cooling (FC) and air cooling (AC)) were used; in this paper, however, results on furnace cooling are reviewed. A phase transformation from fully martensitic $\alpha'$ structure to a microstructure that contained a mixture of $\alpha'$, $\alpha$, and $\beta$ phase with retained prior $\beta$ grains was reported for sample that was heat treated to 705 °C. At 850 °C, the sample was characterized of a microstructure that consisted of needle-like $\alpha'$ and lamella $\alpha + \beta$ phases in a basket weave structure, while, at 940 °C, the authors reported a microstructure that consisted of mixture of $\alpha$ and $\beta$ phases, with the columnar $\beta$ grains no longer visible. This means that a full transformation was observed beginning at 940 °C. Heat treatment at 1015 °C produced a microstructure that contained large $\alpha$-colonies. The observed $\alpha$-phase were transformed into equiaxial grains. A full transformation of martensitic structure into a lamellar $\alpha$ structure in the $\beta$matrix was achieved. Finally, 1050 °C produced a microstructure that was composed of a mixture of lamellar $\alpha + \beta$, with the $\alpha$-colonies forming into a basket weave structure and nucleating at the grain boundaries of big $\beta$ grains.

Finally, Zhang et al. [12] conducted heat treatment on Ti6Al4V samples that were produced with a laser power of 200 W, layer thickness of 30 μm, and scanning speed of 80 mm/s from SLM machine. Samples were heat treated to temperatures of 800 °C and 900 °C, independently, while holding for 2 h (residence time) and followed by cooling in air (AC) and inside the furnace (FC). This study reported a microstructure that contained a high-volume fraction of $\alpha$ and $\beta$ phase for both heating temperatures. In addition, a grain growth for the $\alpha$ phase was reported at 800 °C. Although these reviewed studies show that heat treatment can be used to obtain various type of microstructures which are obviously characterized by different mechanical properties [3,12,14–16], a broader understanding into the effects of all these heat treatment parameters, concurrently, is yet not fully understood. This study focused on investigating the effect of heat treatment parameters (temperature, cooling method, and residence time) on the microstructure and hardness properties of Ti6Al4V parts produced from a high-speed SLM.

## 2. Experimental Procedure

### 2.1. Materials

Spherical grade 5, pre-alloyed Ti6Al4V powder with particle size distribution in the range 20–60 μm was used as a deposition material during printing of the samples. The powder was produced and supplied by TLS Technik GMbH & Co (Bitterfeld, Germany) and was used as received. SEM image of the Ti6Al4V powder is shown in Figure 1.

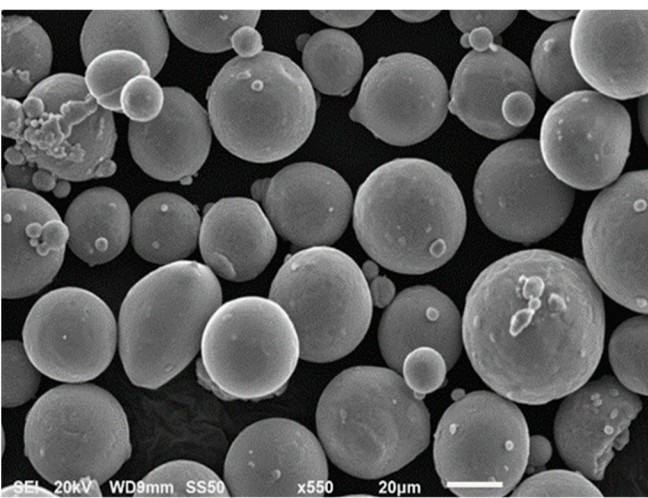

**Figure 1.** SEM image of the pre-alloyed Ti6Al4V Powder.

Figure 1 shows that the powder particles were spherical in shape, with no porosity and surface roughness observed. The samples were printed on an acetone rinsed Ti6Al4V base plate, which was used as a substrate.

### 2.2. Methods

All test samples were manufactured from the Aeroswift 3D printing SLM machine that is available in Pretoria, South Africa. The Aeroswift machine was equipped with a high power 5 kW single Ytteribium laser, with a wavelength of 1076 nm, and a build platform with a volume of 2000 mm × 600 mm × 600 mm. The powder was dried at 120 °C for 2 h in order to remove the moisture before it was processed. Moreover, before the actual printing of the samples could commence, the oxygen content inside the building chamber was reduced to 300 ppm by purging with inert argon gas with purity of 99.99%, and the base plate was pre-heated to a temperature of 200 °C. It is understood that this temperature could help with the reduction of thermal stresses build-up during printing. Post-sample manufacturing, a wire cutter was used to remove the samples from the base plate.

Samples were cut into smaller specimens using a Struers Labotom-5 cutting machine (manufactured by Struers Company, Cleveland, TN, USA) with a high-quality titanium cut-off wheel (20 S25) for metallurgic preparations and heat treatment. All test samples were heat-treated in a horizontal Lenton LFT model furnace (manufactured by Lenton Company, Randburg, South Africa) with an internal tube diameter of 1 m. The image of the heat treatment furnace is shown in Figure 2.

During heat treatment, oxygen concentration inside the furnace was reduced to 200 ppm by purging with argon gas. Gas was kept flowing continuously at a gas rate of 0.8 litres per minute during heat treatment so to avoid oxidation of the Ti6Al4V samples. The furnace is equipped with a K-type thermocouple that was used to measure the temperature inside the furnace during heat treatment. To evaluate the influence of heat treatment temperature, the samples were heat treated to temperatures of 700 °C, 950 °C and 1000 °C, respectively, at the heating rate of 4 °C/min for 2 h and cooled using furnace cooled (FC) method. During heat treatment, samples were placed on a hanging wire, which served as a sample holder.

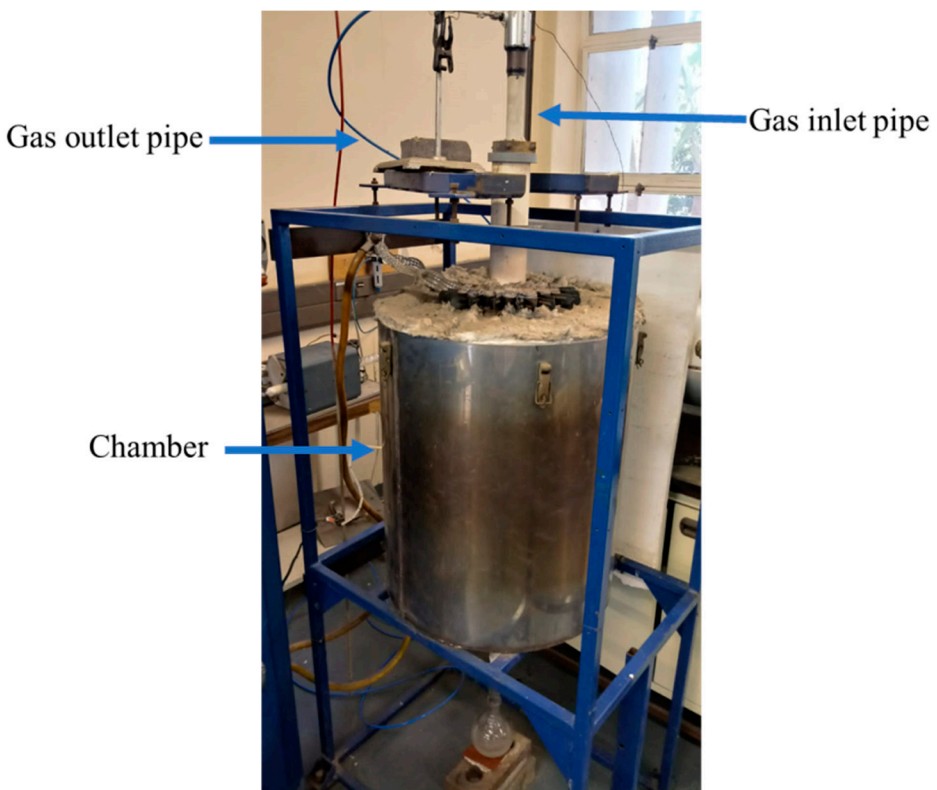

**Figure 2.** Heat treatment furnace available at the University of Pretoria, South Africa.

To evaluate the influence of cooling method, all the test samples were heat-treated to a temperature of 1000 °C at a heating rate of 4 °C/min and held for 2 h and cooled using three different methods, which are water quenching (WQ), air cooling (AC), and furnace cooling (FC). The WQ samples were immersed in a beaker full of tap water, while AC were left-out in an open to cool at room temperature. For the evaluation of residence time, samples were heat treated to 1000 °C, while holding time was varied from 2 h, 4 h, and 8 h, respectively, followed with FC.

The as-built and heat-treated samples were then mounted using an AMP 50 automatic mounting press machine (Beijing, China). AKA Resin Phenolic SEM black conductive resin was used to mount all specimens. Mount specimens were mechanically ground with Struers Tetrapol-25 grinding and polishing machine (manufactured by Struers Company, Cleveland, TN, USA). Silicon carbide (SiC) grinding papers with grit sizes of 80, 320, 1200, and 4000 were used for grinding the specimens, which were later surface finished by polishing to a mirror finish using Diapro MD-Mol 3 μm diamond suspension and colloidal silica 0.04 μm OP-S suspension for 3 min. The specimens were etched with Kroll's reagents, a solution containing 100 mL $H_2O$, 1–3 mL HF, and 2–6 mL $HNO_3$. Samples were immersed in the etchant for about 10–15 s.

The metallographically prepared and etched specimens were observed for microstructure using an Olympus BX51M Optical microscope (manufactured by Olympus, Melville, NY, USA). To observe full view of the microstructure, microstructural images were taken using various magnifications of 5×, 10×, 20×, and 50×. Hardness measurements were performed using Matsuzawa Seiko Vickers model MHT−1 micro-hardness machine (manufactured in Kawabe, Japan). All measurements were done using a diamond type of indenter and applying a force of 0.3 kgf and dwell time of 10 s. For all samples, three hardness patterns were measured, and the average was calculated as the hardness of the material.

## 3. Results

Obtained result, for both as-built and heat treated samples, are reported in this section.

### 3.1. Microstructures

3.1.1. Influence of Temperature

The samples were heat treated to 750 °C, 950 °C, and 1000 °C for 2 h, followed with FC. The microstructures of the as-built and heat-treated samples are shown in Figure 3. For clear microstructural observation, the zoom in areas (on the microstructure) are indicated by red circles on all the microstructures.

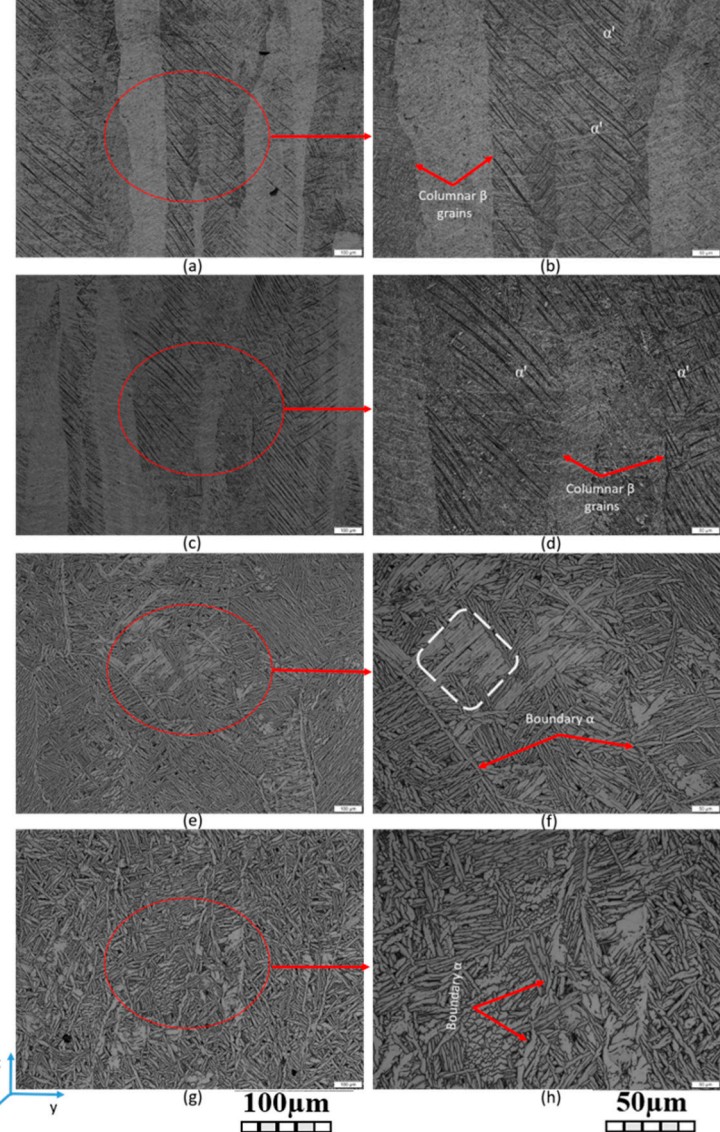

**Figure 3.** Optical microstructures of the samples: (**a**) As-built; (**b**) as-built zoom in; (**c**) 700 °C; (**d**) 700 °C zoom in; (**e**) 950 °C; (**f**) 950 °C zoom in; (**g**) 1000 °C; (**h**) 1000 °C zoom in.

The as-built sample revealed large columnar β grain that were in parallel to the built direction (z-axis), as shown in Figure 3a. These columnar grains are common on the microstructure of SLM processed Ti6Al4V [12,15] because they result from partial re-melting of the previously built layers as a new layer is scanned by the laser beam [1,17]. Inside these large columnar β grains, a fine needle-like martensite α' phase was observed, as shown in Figure 3b. This martensite α' phase is metastable and is formed by high heating and cooling rates experienced by the built samples during the SLM process [9,18].

Heat treatment at 700 °C did not initiate any observable microstructural change, as can be seen in Figure 3c. This was because the microstructure of as-built and 700 °C heat treated samples were observed to be similar in terms of phases and grain orientation. Temperature

of 700 °C is considered a low temperature that can be used to relieve residual thermal stress [8,14,19]. Ter Haar and Becker [11] categorized 700 °C as temperature in the low heat treatment strategy and further recommended it for stress relief purposes. The columnar β grains, together with the martensite α′ phase, were clearly visible after heat treatment at 700 °C, as shown in Figure 3d.

A clear microstructural transformation was observed after heat treatment at 950 °C. The metastable martensite α′ structure was fully transformed into more stable lamella α + β morphology. The observed phase transformation resulted in the changing of the columnar β grains into equiaxed shape, as can be seen in Figure 4e. This implies that 950 °C was able to fully transform the metastable martensite α′ phase. This observation was expected because 950 °C is a temperature slightly below the β-transus temperature for Ti6Al4V. Figure 3f shows that a process of grains growth (area marked in white in Figure 3f) was taking place, with more α-phase growing on the boundaries of the β grain. A similar observation of large amount of grain growth was observed after heat treatment of Ti6Al4V at 950 °C for 1 h [20].

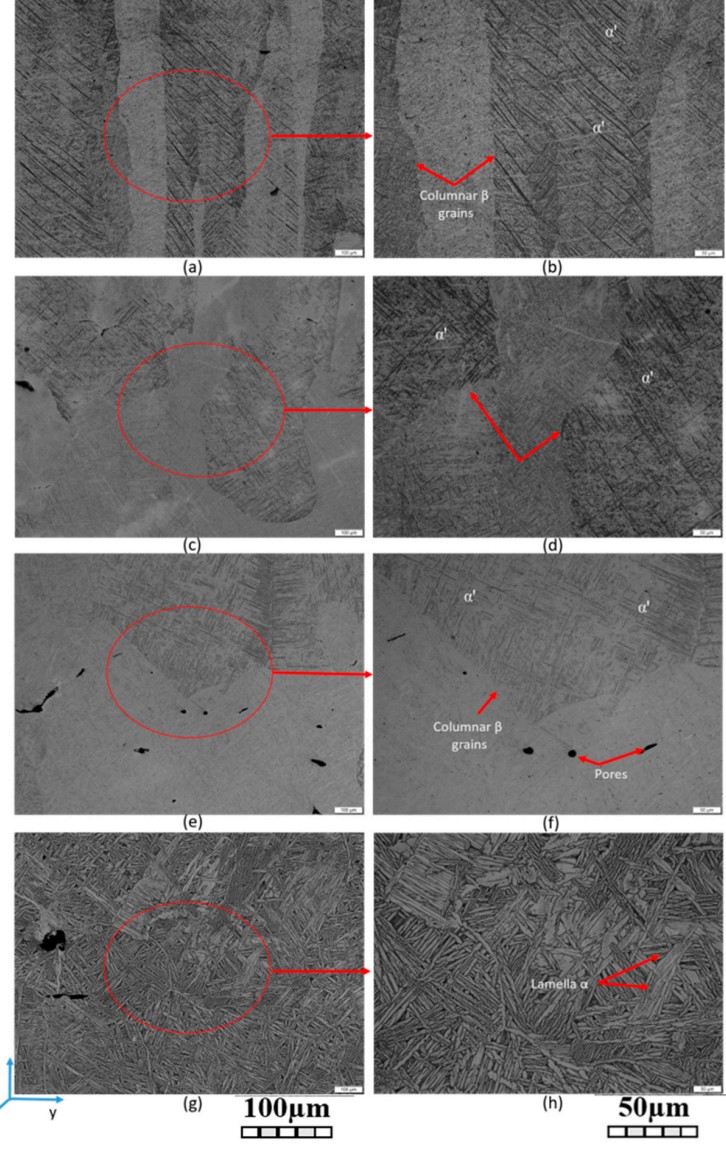

**Figure 4.** Optical microstructures of the samples: (**a**) As-built; (**b**) as-built zoom in; (**c**) water quenching (WQ); (**d**) WQ zoom in; (**e**) air cooling (AC); (**f**) AC zoom in; (**g**) furnace cooling (FC); (**h**) FC zoom in.

Heat treatment at 1000 °C resulted in growth of the lamella $\alpha + \beta$ phases, as can be seen in Figure 3g. This is emphasized more by growth of the $\alpha$-phase on the grain boundaries of the $\beta$ grain, as shown in Figure 3h. The grain growth was associated with the 1000 °C because it is a temperature above $\beta$-transus of Ti6Al4V. According to the phase diagram of Ti6Al4V, heat treating above $\beta$-transus transforms all the phases into $\beta$-phase, which results in growth of the $\beta$-phase. Since 1000 °C is above 950 °C, heat treatment at 1000 °C also resulted in complete transformation of the martensite $\alpha'$ phase into stable lamella $\alpha + \beta$ phases. The same reason accounted for the disappearance of the large columnar $\beta$ grains on the microstructure of the 1000 °C heat-treated sample (Figure 3g).

### 3.1.2. Influence of Cooling Method

The samples were heat treated to 1000 °C for 2 h, followed by cooling with water quenching (WQ), air cooling (AC), and furnace cooling (FC). Figure 4 shows the microstructure of the as-built and heat-treated samples. For clear microstructural observation, the zoom in areas are indicated by red circles on all the microstructures.

Cooling with WQ resulted in the introduction of a newly formed martensite $\alpha'$ phase, which was distributed throughout the entire sample, as shown in Figure 4c. In other word, WQ resulted in the formation of a fully martensitic $\alpha'$ microstructure. Obviously, this was due to the high cooling rate linked with the WQ method [21]. The large columnar $\beta$ grains were observed on the microstructure of the sample that was WQ. Similarly, the microstructure of the sample that was AC also showed a fully martensitic $\alpha'$ structure as presented in Figure 4d. However, the type of martensite $\alpha'$ phase introduced by the AC was finer compared to the one on the as-built and WQ samples. A similar type of finer microstructure was obtained by Matthys [20], after WQ from 870 °C. This martensite $\alpha'$ phase resulted from the high cooling rate associated with the AC method [21].

The microstructure of the FC sample showed different characteristics compared to the as-built, WQ and AC samples. The FC method resulted in the elimination of the undesired metastable martensite $\alpha'$ phase, with the microstructure taking a fully lamella $\alpha + \beta$ morphology as shown in Figure 4h. Most importantly, FC leads to grain growth. The grain growth was due to the slow cooling rates offered by the FC method [21]. A basket weave type of $\alpha + \beta$ morphology was observed on the microstructure of this sample.

### 3.1.3. Influence of Residence Time

The study of residence time was conducted by heat treating the sample to 1000 °C for various residence times of 2 h, 4 h, and 8 h. The microstructures of the sample before and after heat treatment are represented in Figure 5. For clear microstructural observation, the zoom in areas are indicated by red circles on all the microstructures.

The large columnar $\beta$ grains were no longer visible after heat treatment for 2 h, as can be seen in Figure 5c. In addition, heat treatment for 4 h transformed the microstructure into fully $\alpha + \beta$. This implies that 4 h residence time was sufficient to allow full transformation of the martensite $\alpha'$ phase to occur, while allowing the lamella $\alpha + \beta$ grains to grow. The evidence of grain growth was observed more on grain boundaries of the $\beta$ grains as shown in Figure 5d. Heat treatment for 4 h resulted in more grain growth, with formation of large $\alpha$-colonies in a basket weave morphology as presented in Figure 5e.

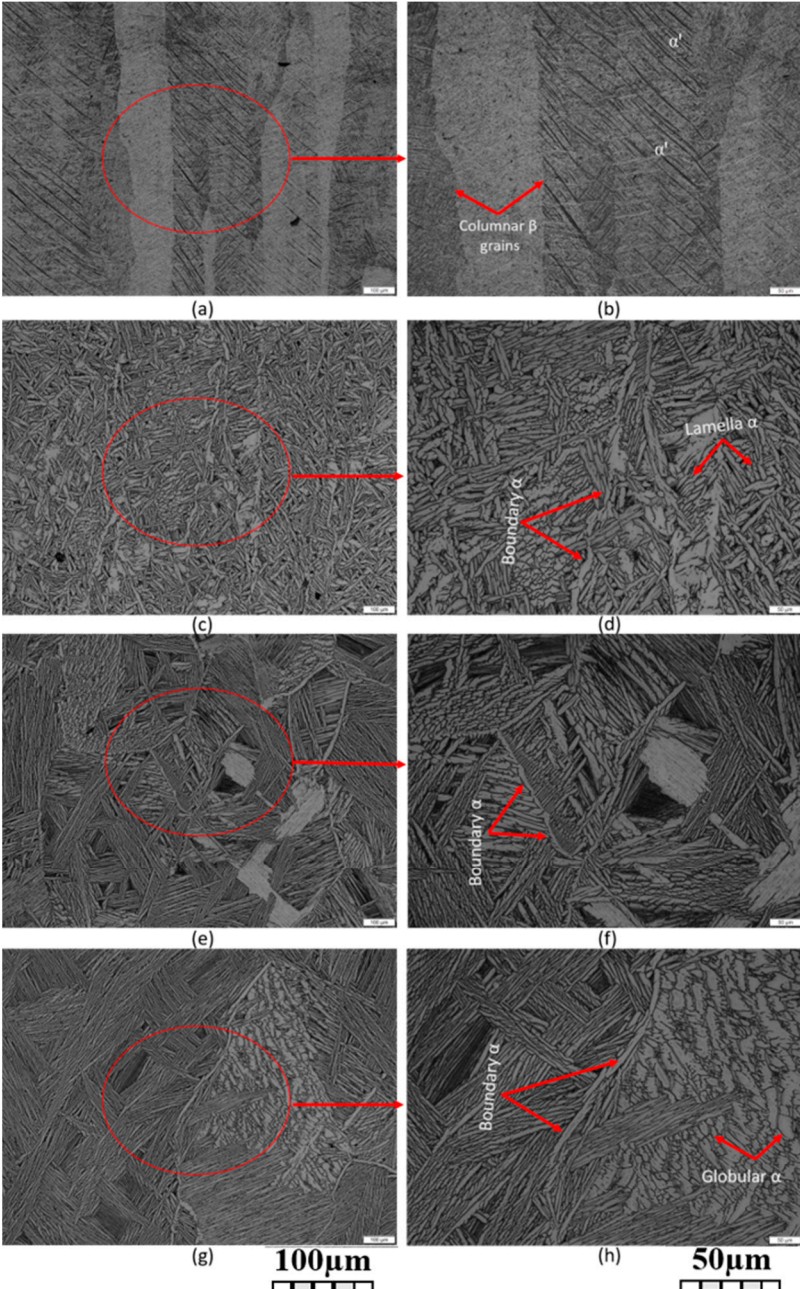

**Figure 5.** Optical microstructures of the samples: (**a**) As-built; (**b**) as-built zoom in; (**c**) 2 h; (**d**) 2 h zoom in; (**e**) 4 h; (**f**) 4 h zoom in; (**g**) 8 h; (**h**) 8 h zoom in.

Heat treatment for 8 h resulted in the thickening of the $\alpha$-phase at the grain boundaries, as shown in Figure 5h. Furthermore, heat treatment for 8 h initiated the process of $\alpha$-phase globularization, as depicted in Figure 5h. Since 8 h is regarded as longer residence time [16], globularization of the $\alpha$-phases formed due to boundary splitting of the $\alpha$-phase through thickening of the lamella $\alpha$-phase [22]. In addition, the globularization of the phases originated at the boundaries of the $\beta$ grains, as can be seen in Figure 5h. This was because of the $\alpha$-phases at the grain boundaries, which are shorter than the other lamella $\alpha$-phases, and, therefore, tend to globularize quicker. The $\alpha$-colonies were clearly visible after heat treatment for 8 h (Figure 5g).

*3.2. Hardness*

3.2.1. Influence of Temperature

Figure 6 and Table 1 present the graph showing hardness profiles and average hardness of the as-built and heat-treated samples, respectively.

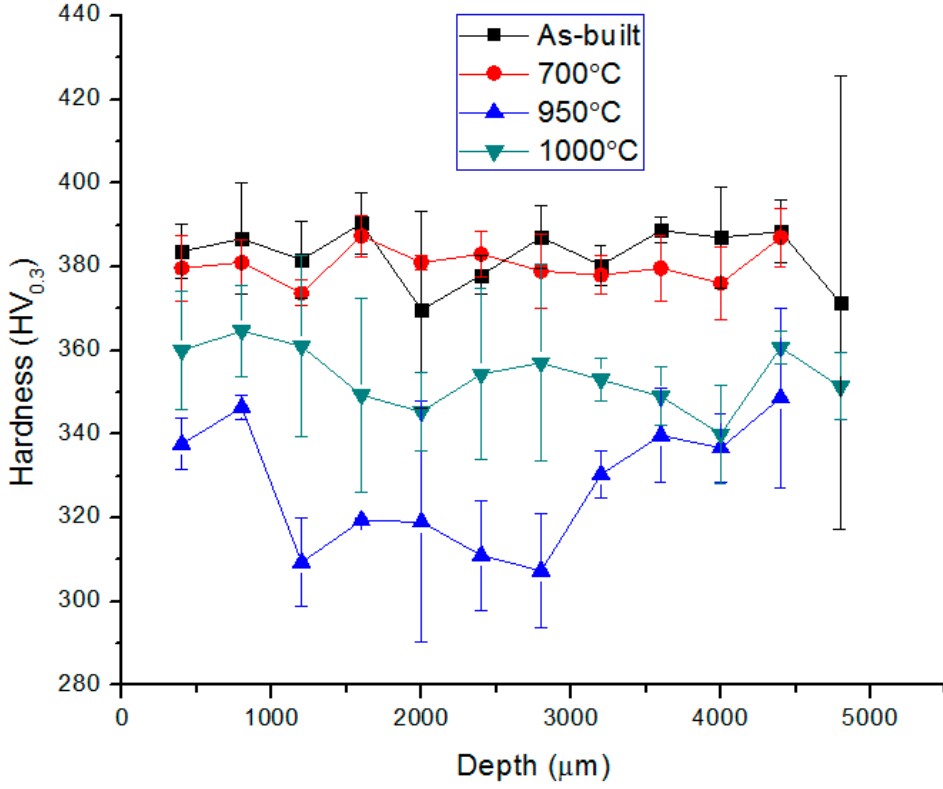

**Figure 6.** Hardness profiles of the as-built and heat-treated samples.

**Table 1.** Average hardness of the as-built and heat-treated samples.

| Samples | As-Built | 700 °C | 950 °C | 1000 °C |
|---|---|---|---|---|
| Hardness (HV) | 383 ± 13 | 380 ± 6 | 327 ± 11 | 358 ± 14 |

The as-built and the 700 °C heat treated samples showed similar hardness profiles (Figure 6). In addition, these two samples gave the highest hardness profiles as can be seen in Figure 6. The similarity in hardness profiles of these samples was attributed to the similarity in their microstructures. Consequently, these two samples gave the highest similar average hardness values as presented in Table 1. The high hardness on these latter samples was attributed to the presence of the martensite $\alpha'$ phase that was observed on microstructures of the samples (Figure 3b,c) [10,12].

Heat treatment at 950 °C and 1000 °C resulted in a visible reduction in hardness (Figure 6). However, 950 °C sample gave the lowest hardness profile and average hardness of 327 ± 11 HV, as can be seen in Figure 6 and Table 1. The decrease in hardness was associated with the transformation of the metastable martensite $\alpha'$ phase into stable $\alpha + \beta$ phases. Even though 950 °C and 1000 °C reduced the hardness, an increase in hardness was observed after heat treatment at 1000 °C, as depicted in Table 1. According to the Hall-Petch relation, the harness of the material is expected to increase upon grain growth. Therefore, this observation was not expected since grain growth was observed on the microstructure of the 1000 °C heat treated sample (Figure 3h) [23].

### 3.2.2. Influence of Cooling Method

The graph showing hardness profiles and average hardness of the as-built and heat-treated samples are shown in Figure 7 and Table 2, respectively.

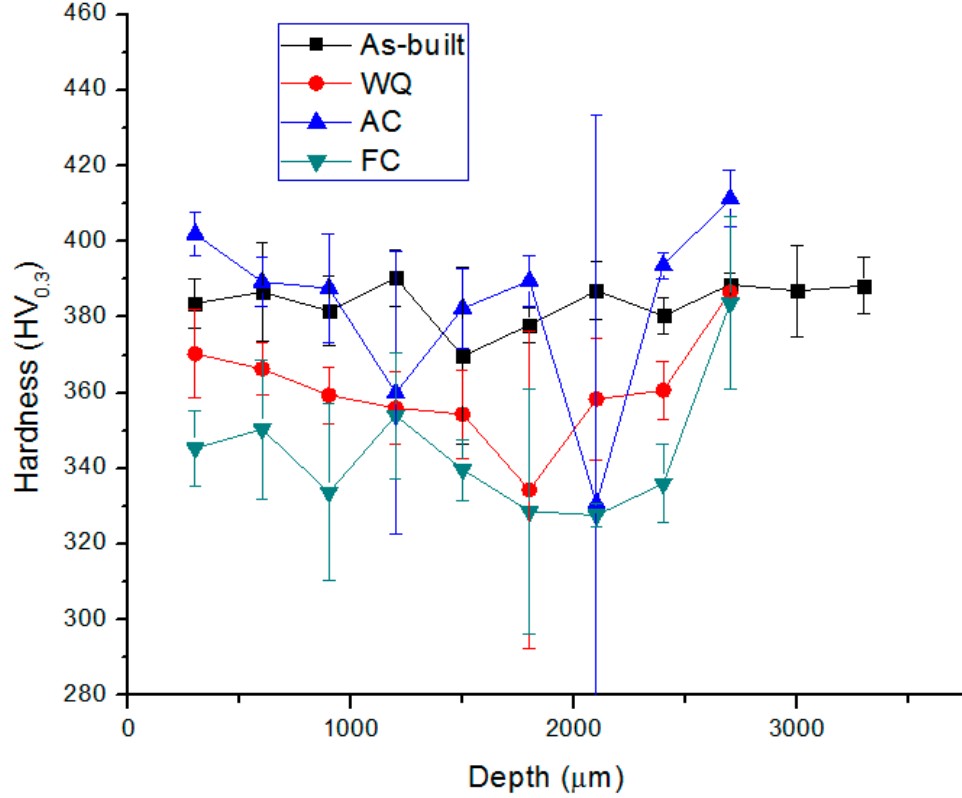

**Figure 7.** Hardness profiles of the as-built and heat-treated samples.

**Table 2.** Average hardness of the as-built and heat-treated samples.

| Samples | As-built | WQ | AC | FC |
|---|---|---|---|---|
| Hardness (HV) | 383 ± 13 | 361 ± 13 | 383 ± 23 | 344 ± 16 |

Figure 7 shows that as-built and AC samples gave the highest hardness profiles. Moreover, the as-built and AC sample gave the same average hardness value of 383 HV as can be seen in Table 2. This observation was expected since both cooling methods produced fully $\alpha'$ structures. A decrease in hardness was observed on the WQ sample. The decrease hardness on this sample was not expected since this cooling method also produced a fully martensitic $\alpha'$ structure, and there was no phase transformation that was observed. A major decrease in hardness was observed on the FC sample, as presented in Figure 7 and Table 2. The decrease in hardness was attributed to the transformation of the martensite $\alpha'$ structure into lamella $\alpha + \beta$ structure.

### 3.2.3. Influence of Residence Time

Figure 8 and Table 2 shows the hardness profiles and average hardness of the as-built and heat-treated samples.

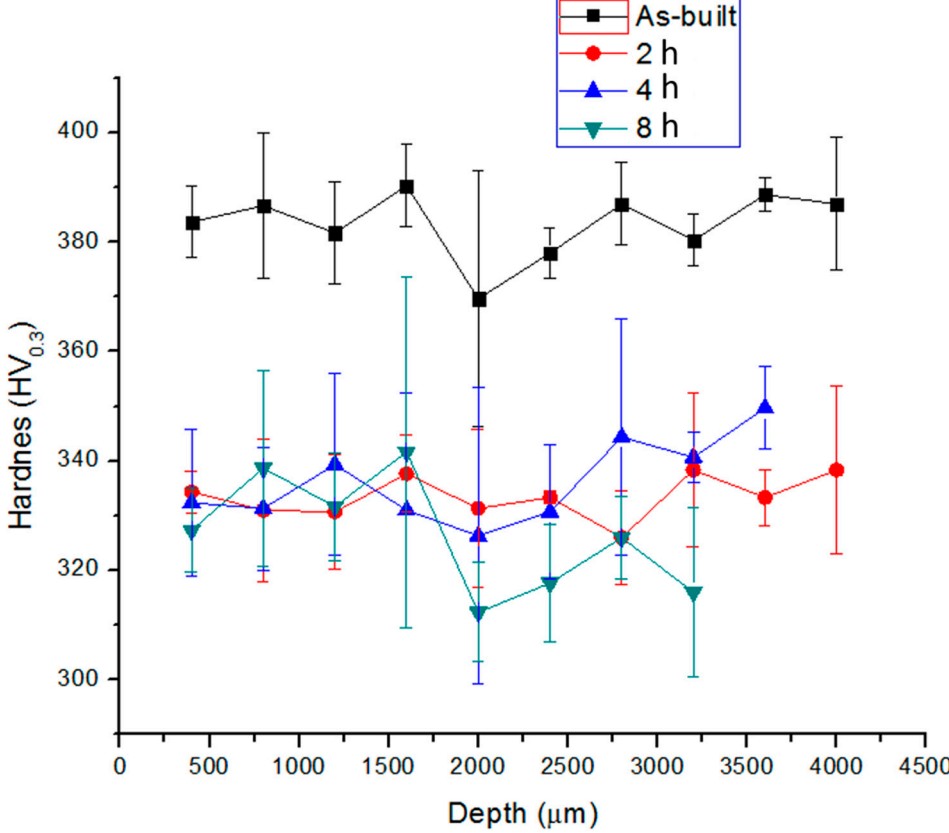

**Figure 8.** Hardness profiles of the as-built and heat-treated samples.

As anticipated, the as-built showed the highest hardness profile due to the martensitic $\alpha'$ phase, as presented in Figure 8 and Table 3. It can be clearly seen from Figure 8 that all the heat treatment for all the residence times resulted in a major decrease in hardness with the 8 h heat treated sample showing the lowest hardness (Table 3). Obviously, this decrease in hardness was attributed to the transformation of the martensite $\alpha'$ phase into stable $\alpha + \beta$ phases. Most importantly, Table 3 shows that heat treatment for all the residence times (2 h, 4 h, 8 h) resulted in similar average hardness results.

**Table 3.** Average hardness of the as-built and heat-treated samples.

| Samples | As-built | 2 h | 4 h | 8 h |
|---|---|---|---|---|
| Hardness (HV) | $383 \pm 13$ | $333 \pm 9$ | $336 \pm 15$ | $327 \pm 14$ |

## 4. Discussion

Heat treatment was employed on SLM Ti6Al4V produced specimens in order to improve their microstructure and hardness properties. The effect of temperature, cooling method and residence time on microstructure and hardness was evaluated. The obtained results are reported in Section 3 and are now discussed in this section.

### 4.1. Temperature Analysis

It is well known that SLM Ti6Al4V is recognized by fully martensitic $\alpha'$ structure due to rapid heating and cooling; hence, post-processing heat treatment is necessary to improve the microstructure and mechanical properties. This study has shown that heat treatment temperature plays an important role in improving the microstructure and hardness characteristics of SLM Ti6Al4V components. An improved microstructure will be characterized by improvement of other mechanical properties, such as strength and elongation (ductility). Both the microstructure of the as-built (Figure 3a) and 700 °C

(Figure 3c) samples showed similar microstructure, with the presence of martensitic α′ phase aligned within large columnar β grains. The similarity in the microstructure of these two samples was confirmed by similar high hardness of 383 ± 13 HV and 380 ± 6 HV for as built and 700 °C sample, respectively, due to the presence of the martensitic α′ phase. These results indicated that 700 °C does not have any observable microstructural change on high speed SLM produced Ti6Al4V components, and that the samples that underwent heat treatment at this temperature would have compromised mechanical properties. Intuitively, this temperature may be necessarily only for residual stress relieving. Similar observation was made by several other authors after heat treatment at temperatures below 730 °C [1,8,14,20,24].

At 950 °C and 1000 °C the samples showed a full transformation of the martensite α′ microstructure into lamella α + β microstructure (Figure 3e,f). This transformation was followed by a major decrease in hardness to 327 ± 11 HV and 358 ± 14 HV for the 950 °C and 1000 °C heat treated samples, respectively. This type of microstructural transformation is characterized by improved ductility of Ti6Al4V components [2,24]. According to Simonelli et al. [2], a lamella microstructure promotes slip transfers between α and β-phases which promote increased dislocation movement, this then result in a sample having an improved ductility. This implies that the transformation into lamella α + β structure and reduced hardness that resulted from heat treatment at 950 °C and 1000 °C improved the microstructure of high speed SLM produced Ti6Al4V components.

While 950 °C and 1000 °C had the same microstructure, it should be noted that the difference in hardness can be attributed to grain formation and size coarsening or refinement, which are known to affect properties, such as strength and hardness. Grain growth was observed to initiate at 950 °C (Figure 3f) and was sustained above this temperature as observed by the growth in the lamellar of the α + β phases reported in Figure 3g. This occurrence resulted in an increase in hardness. This observation did not follow the Hall-Petch relationship [23], which clearly indicate that the hardness of a material is inversely proportional to the size of the grain. Overall, it can be said that both temperatures (950 °C and 1000 °C) were significant in transforming the unwanted martensitic α′ phase into desired α + β phase microstructure. Maybe the grain growth that was observed at 1000 °C was insignificant; hence, no variation in hardness was reported. This, therefore, prompted further investigation on the effects of 1000 °C on the properties of Ti6Al4V SLM built samples, as reported in Sections 4.2 and 4.3, on method of cooling and residence time, respectively.

### 4.2. Cooling Method Analysis

In view of cooling method, it has been already shown that the martensite α′ phase is unwanted and must be avoided due to the characteristics it possesses. WQ and AC methods resulted in the formation of a new martensite α′ structure. A study by Lamirand et al. [21] classified WQ and high cooling and AC as intermediate cooling. This newly formed phase is associated with high cooling rates as is with these two cooling methods. The newly formed martensite α′ phase is typically thinner and finer than the one on the as-built samples [1,25]. A similar type of microstructure was reported by Jovanovich et al. [26] after WQ Ti6Al4V samples from a temperature of 1010 °C. The WQ method produced a coarser type of martensite α′ phase (Figure 4d), while the AC method produced a finer type of martensite α′ phase (Figure 5f). Due to the newly formed martensite α′ phase, the as-built and AC samples gave similar hardness results of 383 ± 13 HV and 383 ± 23 HV, respectively. A decrease in hardness to 361 ± 13 HV was observed on the WQ sample because of the martensitic α′ structure that was observed on the microstructure of this sample (Figure 4c). The observed difference in hardness for AC and WQ can be attributed to grain type that formed. WQ was sustained with coarsening while AC had a finer newly formed martensite α′ phase.

The FC method was able to eliminate the unwanted martensite α′ phase and produced a fully lamella α + β structure that formed into a basket weave morphology (Figure 4g). The resulting lamella α + β microstructure was characterized by a decrease in hardness to

344 ± 16 HV. Since a fully lamella α + β structure is characterized by improved ductility [2,20], the above-mentioned observations implied that the FC method was the only cooling method that was able to improve the microstructure of high speed SLM Ti6Al4V parts and was tested further and the results are reported in Section 4.3.

*4.3. Residence Time Analysis*

As shown in Figure 5c,e,g, all samples that were heat-treated to 1000 °C at different residence time (2 h, 4 h, and 8 h) were able to dissolve and eliminate the formed martensitic α′ phase. The samples that were heat-treated at 1000 °C for 2 h and 4 h produced similar lamella α + β microstructures. These formed lamella α + β microstructures had similar hardness decrease, which was 333 ± 9 HV and 336 ± 15 HV for 2 h and 4 h samples, respectively.

A sample that was heat treated at 1000 °C for 8 h showed different microstructural from that of 2 h and 4 h. The microstructure of the 8 h sample showed a mixture of lamella and globular α + β phases. The globular α + β phases formed on the boundaries of the β grain. Obviously, this was due to prolonged residence time (8 h) [22,27]. This observation was a clear indication that the globularization process occurred. The globularization was attributed to the elongated residence time, which resulted in the thickening of the lamella α + β phases, that initiated α-phase boundary splitting [22]. Heat treatment for 8 h gave a hardness value of 327 ± 14 HV, which was similar to the 2 h and 4 h samples. From the above-mentioned results, it can be said that all the residence time studied improved the microstructure of high speed SLM produced Ti6Al4V components, but 8 h would not be an ideal residence time since a grain thickening occurred in the process during heating.

**5. Conclusions**

- In summary, this study undertook to understand the effect of heat treatment temperature on the microstructure and hardness of the SLM built Ti6Al4V samples. Several temperatures were used, and only 1000 °C seemed interesting enough to further characterized how long should a sample be held at temperature during heat treatment and which cooling method would be suitable to obtain an improved microstructure and mechanical properties for better performance. The results conclusively arrive at an indication 4 h should be the optimum residence time when heat-treating Ti6Al4V at 1000 °C since 8 h led to globalization. Moreover, FC should be the preferred method of cooling since WQ and AC led to formation of the new martensite α′ phase which is undesirable. These observations led to the following conclusion: Heat treatment at 950 °C and 1000 °C produce a lamella α + β microstructure.
- WQ and AC methods led to the formation of new martensite α phase.
- 2 h and 4 h should be taken as best residence time when heat treating Ti6Al4V at 1000 °C since a homogeneous improved lamellar α + β microstructure can be obtained without any grain refinement, coarsening, globalization, or formation of an undesirable new phases.

**Author Contributions:** Conceptualization, P.L., B.M. and M.T.; methodology, P.L., B.M., M.T. and K.A.; validation, B.M. and M.T.; investigation, P.L.; data curation, P.L.; writing original draft preparation, P.L., B.M. and M.T.; review and editing, P.L. and M.T.; supervision, B.M., M.T., K.A. and N.M.; project administration, B.M.; funding acquisition, B.M. All authors have read and agreed to the published version of the manuscript.

**Funding:** This research was funded by the Department of Science and Innovation (DSI) of South Africa (SA) through the Collaborative Program in Additive Manufacturing (CPAM), grant number LHEBBDL.

**Informed Consent Statement:** Not applicable.

**Acknowledgments:** Laboratory and equipment support from CSIR National Laser Centre and the University of Pretoria is acknowledged.

**Conflicts of Interest:** The authors declare no conflict of interest.

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
