# Peer review of "Evaluation of Heat Treatment Parameters on Microstructure and Hardness Properties of High-Speed Selective Laser Melted Ti6Al4V"

_metals, doi:10.3390/met11020255_

Round 1
Reviewer 1 Report
Evaluation of Heat Treatment Parameters on Microstructure and Hardness Properties of High Speed Selective Laser Melted Ti6Al4V
In the paper there are some doubts that should be resolved. For this reason, in my opinion, the paper needs a minor revision before it is ready for publication.
- Introduction
The introduction is well articulated but slightly poor, especially given the large number of publications on the subject.
- Methods
- Page 4, line 109 - there is a typo “;,”
- For an easier interpretation of the initial microstructure of the samples it would be advisable to report the parameters of the SLM process.
- For the reviewer's opinion, it would have been better to perform a doe and a statistical analysis to better evaluate the influence of each parameter or the combination of parameters on the microstructure and hardness. Can the authors explain the reason for choosing this method?
Reviewer 2 Report
The manuscript reports the results of experimental research of effect of heat treatment parameters on microstructure and hardness in additive manufactured Ti-6Al-4V alloys via SLM method. The results are new, but the novelty and originality is inadequate. Thus, the referee does not think it is suitable for publication in metals.
Reviewer 3 Report
The authors present influence of heat treatment on microstructure and mechanical properties of high-speed selective laser melted Ti6Al4V. Considering investigated material, the idea of such a study is not new but the manuscript does contain novel results. Based on the obtained results, a detailed description of the influence of the heat treatment parameters (temperature, cooling method, residence time) on microstructure and hardness is presented. The paper is interesting and useful. The presented analysis is logical, and the number of references is sufficient. In general, the presented results are of interest to scientists and engineers. Results of the research are relatively clear, but the manuscript needs a minor revision. The only one minor comment is given:
Scale bars in Fig 3-5 are not visible. Please add new.
Round 2
Reviewer 2 Report
This manuscript has been improved for publication.